Reef-scale trends in Florida Acropora spp. abundance and the effects of population enhancement

Miller Margaret W. margaret.w.miller@noaa.gov 1
Kerr Katryna 2
Williams Dana E. 1 2
1 Southeast Fisheries Science Center, NOAA-National Marine Fisheries Service , Miami , FL , United States
2 Rosenstiel School of Marine and Atmospheric Sciences/Cooperative Institute of Marine and Atmospheric Studies, University of Miami , Miami , FL , United States
Reimer James
Electronic publication date: 2016 Sep 29
Publication date: 2016
Volume: 4
Electronic Location ID: e2523
Received 2016 Jun 7; Accepted 2016 Sep 3
Copyright year: 2016
License: This is an open access article, free of all copyright, made available under the Creative Commons Public Domain Dedication. This work may be freely reproduced, distributed, transmitted, modified, built upon, or otherwise used by anyone for any lawful purpose.
License URL: https://creativecommons.org/publicdomain/zero/1.0/

Keywords: Coral nursery, Bleaching, Spatial analysis, Outplanting, Coral restoration, Florida Keys

Funding: NOAA Coral Reef Conservation Program This study was funded by the NOAA Coral Reef Conservation Program. The funders had no role in study design, data collection and analysis, decision to publish, or preparation of the manuscript.

==============================
Since the listing of Acropora palmata and A. cervicornis under the US Endangered Species Act in 2006, increasing investments have been made in propagation of listed corals (primarily A. cervicornis, A. palmata to a much lesser extent) in offshore coral nurseries and outplanting cultured fragments to reef habitats. This investment is superimposed over a spatiotemporal patchwork of ongoing disturbances (especially storms, thermal bleaching, and disease) as well as the potential for natural population recovery. In 2014 and 2015, we repeated broad scale (>50 ha), low precision Acropora spp. censuses (i.e., direct observation by snorkelers documented via handheld GPS) originally conducted in appropriate reef habitats during 2005–2007 to evaluate the trajectory of local populations and the effect of population enhancement. Over the decade-long study, A. palmata showed a cumulative proportional decline of 0.4 –0.7x in colony density across all sites, despite very low levels of outplanting at some sites. A. cervicornis showed similar proportional declines at sites without outplanting. In contrast, sites that received A. cervicornis outplants showed a dramatic increase in density (over 13x). Indeed, change in A. cervicornis colony density was significantly positively correlated with cumulative numbers of outplants across sites. This study documents a substantive reef-scale benefit of Acropora spp. population enhancement in the Florida Keys, when performed at adequate levels, against a backdrop of ongoing population decline.

Introduction

Caribbean coral reefs are home to two species of fast-growing, habitat-forming species of Acropora spp. corals; staghorn (A. cervicornis) and elkhorn (A. palmata). Both are listed as Critically Endangered by IUCN and threatened under the US Endangered Species Act (ESA). Their endangered status accrues from a litany of factors which have caused extensive mortality combined with inadequate recruitment to sustain populations throughout their range (Acropora Biological Review Team, 2005; Aronson & Precht, 2001; Bright, Williams & Miller, 2013). ESA listing carries a legal mandate to ‘recover’ imperiled species. The Recovery Plan for A. palmata and A. cervicornis (NMFS, 2015) describes the need for ongoing monitoring and evaluation to track the status of populations, as well as the need to curb ongoing threats (e.g., disease, land-based sources of pollution, and thermal stress due to global climate change) and implement proactive population enhancement measures to jumpstart population recovery (NMFS, 2015). Growing effort has been dedicated to implementing population enhancement throughout the Caribbean (Young, Schopmeyer & Lirman, 2012), largely following the ‘coral gardening’ model (Epstein, Bak & Rinkevich, 2003; Rinkevich, 2015).

As Acropora population enhancement effort has grown, substantial management and planning effort has been invested into developing risk-averse strategies. These strategies include: 1. emphasis on in situ (versus land-based) culture; 2. dispersing individual field nursery operations to limit the geographic distance from which source stocks are drawn and propagated fragments are outplanted; and 3. maximizing and tracking the genotypic diversity of cultured stocks. Acropora spp. are propagated via fragmentation from locally-collected stocks in offshore field nurseries, grown to a viable size and then outplanted to reef habitats with the goal of re-creating sustainable population patches which can serve as larval sources to jumpstart population recovery on a broader scale. Common practices and details of implementation are described in Johnson et al. (2011).

Unfortunately, cultured Acropora fragments often behave like their wild counterparts in Caribbean reef communities as they are subject to ongoing chronic and acute stressors, often manifesting substantial mortality in the same pattern as the background population (Miller et al., 2014; Schopmeyer & Lirman, 2015). Critics of population enhancement maintain that potentially high levels of mortality would preclude any long term benefit to population recovery, and that high cost implies that the scale of effect (e.g., area of reef) will remain trivial. Substantial published work has documented the remarkable success of these field nursery culture efforts (Griffin et al., 2012; Lirman et al., 2014; Lirman et al., 2010; Lohr et al., 2015) as well as the short term fate of individual outplanted colonies (Griffin et al., 2015a; Mercado-Molina, Ruiz-Diaz & Sabat, 2015). These evaluations are based on tractable observations and measurements of individual tagged colonies at a few sites over one to a few years. There is a much greater challenge in tracking Acropora spp. colony abundance at the meso-scale (100’s m2 to hectares) due to fragmentation, displacement, and partial mortality. Consequently, there is little information available to aid in evaluating the potential for active population enhancement to ‘move the needle’ in affecting reef-scale population trajectories of Caribbean Acropora spp.

In this study, we used a broad scale, low-precision census technique (direct observation by snorkelers documented via handheld GPS; Devine, Rogers & Loomis, 2005; Walker et al., 2012) to further this evaluation goal by documenting both the long term (2005–2015) reef-scale trajectory of Florida Keys Acropora populations undergoing ongoing acute and chronic disturbances as well as whether these trajectories are influenced by population enhancement effort. The acute disturbances affecting these populations included multiple tropical storms (2005, 2008, 2012), a severe cold thermal event in 2010, mild bleaching in 2011, and a severe warm thermal mass bleaching event in 2014 as well as chronic and substantial effects of predation and disease (Williams & Miller, 2012).

A conservation organization in the upper Florida Keys (Coral Restoration Foundation, CRF) has been propagating and outplanting A. cervicornis since 2003 (substantial numbers since 2011) and A. palmata since 2012 (substantial numbers since 2014, Table S1), although the number of outplants placed on the reef has varied greatly over time according to the factors such as permitting restrictions, damaging storms which required time for recovery of nursery infrastructure and cultured stocks, and funding levels. This sustained effort combined with the availability of historic census information from a range of reef sites in the upper Florida Keys provides a novel opportunity to evaluate potential reef-scale effects of Acropora spp. population enhancement against a backdrop of ongoing chronic and acute disturbances in the reef environment. We compared trajectories of Acropora spp. density at reef areas which had versus had not received population enhancement efforts over appropriate time frames to evaluate the reef-scale effect of enhancement.

Methods

Sites targeted for this study were chosen in 2005, prior to the onset of substantial population enhancement efforts. Habitat maps (as described in Lidz et al., 2006; Marszalek, 1977) were used to identify shallow (<5 m) coral habitat areas in the upper Florida Keys, spanning between Carysfort reef in the north to Pickles reef in the south (i.e., 25.2°N–24.9°N latitude). Targeted reef areas were restricted to less than 5 m depth as observations at deeper depths on snorkel become less reliable. This depth range encompasses the core habitat for A. palmata, though A. cervicornis traditionally occupies a wider depth range. Most, but not all sites were surveyed once in 2005–7, once in 2013 or 14, and once in 2015 (Table S2; Miller, 2008; Williams, 2013). Hence, different numbers of sites are available for different temporal comparisons.

Teams of two or three snorkelers addressed each study site with the intent to observe the entire reef surface via swimming sequential, parallel linear transects. The width of each transect was adjusted according to conditions including depth, relief and water visibility, with the intent that the benthos was thoroughly observed with minimal overlap. In practice, this is very challenging to accomplish and enhanced procedures were implemented as the effort progressed to improve the practical coverage, including the visual delineation of the target area (or subset assigned to an individual snorkeler) with weighted dive flags and the use of compasses and pre-agreed headings (generally following the direction of reef spurs) to maintain parallel tracks. In the early censuses, dive scooters (SeaDoo VS Supercharged) were used, but snorkelers performed surveys predominantly under their own power in 2013–15.

Each snorkeler towed a handheld GPS unit (Garmin GPS72 in 2005–7; Garmin eTrex20 for 2013–15) in a waterproof plastic pouch attached to a floating dive flag. The GPS recorded the ‘track’ traversed by the snorkeler. When an Acropora spp. colony was encountered, the snorkeler recorded a waypoint on the GPS for each, and recorded the species for each waypoint on a field data sheet. In some cases, A. palmata colonies were observed growing in high density patches wherein it was not feasible to demarcate individual colonies. In these cases, the snorkeler would swim around the perimeter of the feature and record waypoints along the outline which were designated on the data sheet as a ‘thicket.’ While it is possible that this qualitative definition may have been applied slightly differently by different observers, the disappearance of known thicket areas (e.g., Grecian Rocks and Watsons reef) at later surveys was verified by multiple observers. The area occupied within this ‘thicket’ outline was calculated in GIS for each survey and was compared at each site over time. We did not document any analogous ‘thickets’ for A. cervicornis.

Figure 1 Illustration of survey and spatial analysis.

Component maps and spatial analyses are illustrated for a single site, Grecian Rocks; a similar sequence of maps was constructed for each site (given in Fig. S1) and temporal comparison. Observed search tracks and waypoint features mapped for each census year (2006 points as stars, 2014 points as asterisks) are given in (A) 2006 with waypoints as stars and (B) 2014 with waypoints as asterisks. A. palmata colony waypoints are depicted in yellow, A. cervicornis colonies in purple, and A. palmata thicktet outline points in red. (C) Merged maps for temporal comparison showing the congruent observed area (determined by GIS intersect of the polygons determined by the search tracks for each year) for both years and the overlayed colony distribution observed in both years. Similar year and temporal comparison maps for 2014–2015 are given in Fig S2.

After each survey was completed, the GPS-recorded track was saved, waypoints and tracks were downloaded to a personal computer, and then exported to a spreadsheet file where the waypoint attributes were entered from the field data sheet. For each study site, maps were created in ArcGIS plotting the colonies and thickets observed for each census year (Figs. 1A–1B; Fig. S1). Lastly, the observational paths followed by the surveyors (i.e., the GPS tracks) were imported to each map to depict the area searched. Using the Minimum Bounding Geometry tool, the minimum area covered by the observational path (observed area) was determined for each year*site map.

Temporal comparisons were made for two intervals: a long-term interval from the periods 2005–07 (early; e.g., Fig. 1A) to 2013–14 (late; Fig. 1B); and a short term interval from 2013–14 (early; Fig. S1A) to 2015 (late; Fig. S1B). For each site, the early and late maps were merged to make temporal comparisons of reef scale density trends (Fig. 1C; Fig. S1C). Each temporal comparison was restricted to congruent observed areas of the reef (i.e., covered by the observational paths in both time points) by clipping the area of comparison to the area of overlap in the observed area for each year using the Intersect tool. If the congruent area consisted of numerous overlapping polygons, then the Merge Polygon tool was used. Areas outlined as A. palmata ‘thicket’ were calculated for each survey and the number of colonies in each thicket area was estimated using a standard density estimate of 1 colony per m2 (based on independent field estimates using fixed area belt transects within the Horseshoe reef thicket over four years yielding a mean of 1.01 ± 0.26 SD colonies per m2; M Miller, 2010, unpublished data). Individual colony waypoints and thicket abundance estimates were summed for each species to obtain the total abundance for each survey year in the overlapping comparison area.

Total colony abundance of each species in each year in the congruent search area of reef was converted to density (total number of colonies observed/congruent observed area of reef (m2)) to compare between time points (Mann–Whitney Rank Sum tests). For temporal comparisons, the proportional change in density between two time points was calculated. This proportional change in density at each site was calculated for the longest interval observed for each site, as well as the pre- (2005–6 versus 2014) and post-bleaching (2014 versus 2015) intervals. Proportional change in density was annualized by dividing by the number of years between surveys and Mann–Whitney Rank Sum tests were used to test for significant difference between the pre- and post-bleaching intervals.

Information on the total number of coral colonies of each species outplanted to each censused reef site by year was provided by staff of the Coral Restoration Foundation (J Levy & K Ripple, pers. comm., 2015; Table S1). CRF is the only organization undertaking large scale Acropora spp. population enhancement in the study area (additional nurseries operate and outplant in different sectors of the Florida Reef Tract). The overall change in mean density over the longest observed interval for each reef (Table S3) was used to correlate the overall impact of population enhancement for A. cervicornis as outplanting has been ongoing for this species since 2008 and this enabled the use of information from all sites (n = 14, a few of which had not been surveyed in 2013 or 14). However, substantial outplanting was only conducted for A. palmata since 2014 (Table S1) so the 2014–2015 interval only was used to correlate with outplanting effect for this species. For each species, we conducted a Mann–Whitney rank sum test comparing proportional change in colony density between the sites which had and sites which had not received outplants. Also, a simple linear regression was performed for each species between the proportional change in colony density and the cumulative number of outplants among all sites.

Results

The total surveyed area for each census ranged from 55 to 77 hectares while the congruent observed area of reef for temporal comparisons within each site ranged from 1.6 to 15.5 hectares (Table 1). Acropora palmata thickets were observed at four sites in 2005–7, two of which had disappeared by 2015 (Fig. 2). At these two sites (Grecian Rocks and Watsons Reef) the aggregation of A. palmata colonies in the thicket area had dwindled to where it was no longer designated a thicket, though a few widely-spaced remnant colonies remained. One of the other two sites with thickets showed approximately half decline in area, whereas the last was approximately stable in area (Fig. 2). Overall, this represents over two thirds loss in total A. palmata thicket area (from 2,229 m2 to 713 m2) among these four sites.

Table 1 Long-term and short-term Acropora spp. changes at sites in the upper Florida Keys.

Summary of congruent observed areas, colony densities, and number of outplants for both species over (A) long term interval prior to 2014 thermal bleaching event and (B) over the 2014–2015 bleaching event, at sites in the upper Florida Keys. Change in density is represented as a proportion of the initial density. Information on numbers of outplants provided by Coral Restoration Foundation, the only organization performing large-scale population enhancement in this region. ‘Early’ and ‘Late’ refer to the first and last survey year, respectively, of the interval for each site. Density expressed as number of colonies per hectare.

			A. cervicornis	A. palmata	
Reef	Years	Congr Area (ha)	#Ac-early	AcDens-early	#Ac-late	AcDens-late	Change AcDens	# Ac Outpl	#Ap-early	ApDens-early	#Ap-late	ApDense-late	Change ApDens	# Ap Outpl	
(A)	
CF	05 & 14	1.6	8	4.9	9	5.6	0.1	370	55	34.0	0	0.0	−1.0	0	
FR	07 & 14	8.0	8	1.0	41	5.1	4.1	682	185	23.2	63	7.9	−0.7	0	
ML	06 & 14	15.5	12	0.8	1331	85.8	109.9	3071	239	15.4	89	5.7	−0.6	0	
NDR	06 & 14	3.9	109	28.3	3	0.8	-1.0	300	74	19.2	27	7.0	−0.6	50	
GR	06 & 14	14.1	42	3.0	39	2.8	−0.1	0	408	29.0	276	19.6	−0.3	0	
WBDR 2	06 & 14	6.4	10	1.6	526	82.3	51.6	1307	0	0.0	0	0.0	0.0	0	
WBDR 1	06 & 14	9.3	172	18.6	448	48.4	1.6	0	6	0.6	0	0.0	-1.0	0	
LG	06 & 13	2.1	1	0.5	8	3.8	7.0	0	320	153.8	270	129.8	−0.2	0	
(B)	
CF	14 & 15	9.4	0	0.0	0	0.0	0.0	815	26	2.8	12	1.3	−0.5	66	
FR	14 & 15	7.3	40	5.5	102	13.9	1.6	0	63	8.6	82	11.2	0.3	230	
ML	14 & 15	13.5	1260	93.2	269	19.9	−0.8	915	89	6.6	82	6.1	−0.1	377	
NDR	14 & 15	3.1	109	34.7	79	25.2	−0.3	388	74	23.6	137	43.6	0.9	170	
GR	14 & 15	9.3	39	4.2	231	24.8	4.9	603	241	25.9	50	5.4	−0.8	0	
WBDR2	14 & 15	5.6	526	93.4	194	34.5	−0.6	0	0	0.0	0	0.0	0	0	
WBDR1	14 & 15	6.2	446	71.7	84	13.5	−0.8	0	0	0.0	0	0.0	0	0	
NNDR	13 & 15	2.6	8	3.1	0	0.0	−1.0	0	34	13.0	88	33.7	1.6	0	
LG	13 & 15	3.5	18	5.1	15	4.3	−0.2	0	60	17.1	28	8.0	−0.5	0	
SI	14 & 15	3.5	0	0.0	0	0.0	0.0	0	106	30.7	35	10.1	−0.7	0	
Notes.

CF Carysfort

FR French

ML Molasses

NDR North Dry Rocks

WBDR1/2 White Bank Dry Rocks north/south

LG Little Grecian

GR Grecian Rocks

NNDR North North Dry Rocks

SI Sand Islandad

Figure 2 Acropora palmata thickets.

Area (m2) of A. palmata thickets (i.e., high density aggregations for which mapping individual colonies was deemed infeasible) at four sites over time. Horseshoe was surveyed in both 2005 and 2007 so the point for this time period is a mean of these two. Thickets dropping to zero area likely still contained remnant colonies, but at lower densities such that individual colonies could be mapped (see text for details on methods).

Table 2 Overall changes in colony density.

Cumulative changes in Acropora spp. colony density over the full study duration (2005–07 versus 2014–15) pooled among sites with and without outplanting for each species. Sites and specific durations for each given in Table S2. Density expressed as number of colonies per hectare.

	Total Congr area (ha)	# colonies early	Density early	# colonies late	Density late	Change in density	
A. cervicornis	
With outplants (n = 7 sites)	50.8	93	1.8	1356	26.7	13.6	
Without outplants (n = 7 sites)	33.2	227	6.8	144	4.3	−0.4	
A. palmata	
With outplants (n = 5 sites)	33.8	620	18.3	369	10.9	−0.4	
Without outplants (n = 9 sites)	50.2	671	13.4	178	3.5	−0.7	

When considering the full study duration, both species showed a negative trajectory in the absence of outplanting (40% decline in density for A. cervicornis, 70% decline for A. palmata, pooled among 7 or 9 sites respectively; Table 2). When considering trends between the pre- and post-bleaching intervals, A. cervicornis showed a substantial annualized increase in density when averaged across all sites from 2005 to 2014 (n = 8, (Table 1A) and Fig. 3A)) with the most dramatic changes occurring at sites receiving outplants (Table 1A). A. cervicornis density increased only slightly on average between 2014 and 2015 (co-incident with a mass thermal bleaching event) yielding a significant difference in annualized density change between the two intervals (Mann–Whitney Rank Sum Test p = 0.037; Fig. 3A). Meanwhile, A. palmata showed much smaller proportional changes in density (corresponding with many fewer total outplants; Table 1 and Fig. 3). While the average trend was substantially negative in the 2005–14 interval and essentially stable in the 2014–15 (bleaching) interval (Fig. 3), this difference was not statistically significant (t-test, p = 0.824).

Figure 3 Acropora spp. change in density

Annualized proportional change in colony density (i.e., proportion change from Table 1 divided by the number of years in the observed interval; mean plus 1 SE) for Acropora cervicornis (A) and Acropora palmata (B) during two time intervals. Diamonds (right y-axis) show the mean number of fragments of each species outplanted per year over the same intervals. Note the differences in axis scales.

To specifically evaluate the hypothesis that outplanting effort had a significant, landscape-scale effect on colony density, we performed two separate tests. For A. cervicornis these tests were applied to the full interval of observation at each site (2005–2015, n = 14 sites, Table S3) whereas for A. palmata, substantial enhancement effort has only occurred since 2014 so the 2014–2015 interval was used. A Mann–Whitney U-test indicated that sites receiving A. cervicornis outplants had significantly different change in density than those that did not (p = 0.002). However, no significant difference occurred for A. palmata (corresponding to a much smaller cumulative number of outplants, Table 1). Simple linear regression showed a strong and highly significant relationship between change in A. cervicornis colony density and cumulative number of outplants among sites (Fig. 4A). The similar regression for A. palmata for the 2014–2015 interval when outplanting (as well as the bleaching event) occurred showed no significant relationship (Fig. 4B).

Figure 4 Acropora spp change in colony density with population enhancement.

Scatterplot showing linear regressions for proportional change in colony density relative to the cumulative number of outplants for (A) Acropora cervicornis (full interval of observation, n = 14 sites) and (B) Acropora palmata. Population enhancement has only occurred for A. palmata since 2014, so (B) shows proportional change in density for this species from 2014–2015 at n = 9 sites (regression is not significant).

Discussion

This study is not intended to provide an overall cost:benefit for Acropora spp. population enhancement as many details of stock collection, propagation, and short term colony-scale outplant success have been previously documented (Griffin et al., 2012; Griffin et al., 2015b; Johnson et al., 2011; Lirman et al., 2014; Lirman et al., 2010; Lohr et al., 2015; Mercado-Molina, Ruiz-Diaz & Sabat, 2015; Young, Schopmeyer & Lirman, 2012). Rather, we sought, via a low precision but large scale census approach, to determine if reef-scale effects of population enhancement efforts could be discerned. We performed repeated censuses over multiple reef sites over a decadal time frame in which both extensive population enhancement effort and an acute thermal disturbance (along with several lesser disturbances) occurred. Thus, this approach was designed to detect large changes at a large spatial scale.

Our surface-based observation method restricted censused areas to generally less than 5 m depth. The depth of outplants at each site is not consistently documented, though some were likely placed deeper than 5 m depths and missed in our surveys, rather than dead. Although the historic core habitat of A. cervicornis likely extended deeper than 5 m depth, current known distribution of A. cervicornis in the Keys is predominated by nearshore (shallower) habitats (Miller et al., 2008) in contrast to the deeper fore-reef habitats historically described for this species. Thus, although extensive A. cervicornis distribution in deeper areas not covered by our study is possible, current evidence does not support this in the Florida Keys.

Much greater overall enhancement effort (∼ an order of magnitude, Table 1; Table S1) has gone to A. cervicornis in comparison to A. palmata, and this added effort corresponded to a significant landscape scale effect. A. cervicornis density showed a significant and positive relationship with the degree of this enhancement effort across sites over the entire study period (Fig. 4A). However, the acute thermal bleaching event appears to have reduced the impact of outplanting between 2014 and 2015 as more than double the previous annual outplanting effort during that year yielded a much smaller increment of density compared to the earlier interval (Fig. 3A). Indeed, we observed extensive bleaching and mortality of outplants during the 2014 bleaching event in a separate study (MW Miller & DE Williams, 2014, unpublished data). Both the overall densities and the scale of the enhancement effort have been lower for A. palmata (Table 1) which shows a clear pattern of declining density over the recent decade both overall (Table 2 and Fig. 3B) and as represented in the occupation of thickets (Fig. 2). This mostly negative population trend has not been substantively overcome by the small outplanting effort to date and is consistent with results of independent plot-scale studies in the Florida Keys (Sutherland et al., 2016; Williams & Miller, 2012). We are not aware of other published studies of contemporary trends in Florida Keys A. cervicornis populations, although the detrimental effects of individual events on this species are documented (e.g., 2010 winter cold; Kemp et al., 2011). Substantial documentation does exist (Vargas-Angel, Thomas & Hoke, 2003; Walker et al., 2012; D’Antonio, Gilliam & Walker, 2016) of abundant A. cervicornis populations, including extensive thickets, in southeast Florida (over 80 km north of our study area), though little quantitative information on trends in abundance is available in this nearby region.

The precision of our census technique was low, as the challenge of a snorkeler navigating in open ocean as well as variation in depth, visibility, and likely individual observer variation yielded less than perfect observational coverage and detection of colonies that were present. However, we implemented improved field techniques over time which improved the operational coverage of the area surveyed (e.g., the deployment of surface markers to delineate the survey area for each surveyor and the use of compasses; compare coverage of tracks in Figs. 1A vs 1B; additional tracks in Fig. S2). Thus, results suggesting overall decline in densities are conservative as we expect our observational detection was improved in later years. Also, this technique allowed us to evaluate population trends at a hectare scale. More resolved techniques such as photo mosaics provide a much more precise assessment technique, but are still only applicable at meso-scales (hundreds of m2; Lirman et al., 2007).

The substantial loss of A. palmata thicket area is a particularly concerning result. Acropora thickets are understood to have been the typical configuration on Caribbean reefs prior to the drastic decline of these species starting in the late 1970’s (Gladfelter, 1982; Goreau, 1959; Jaap, 1984) and are functionally important in terms of providing structural habitat both for other reef inhabitants and to facilitate fragment retention (i.e., successful asexual reproduction) for the coral itself. This importance is reflected in the fact that the area of thickets (not just population abundance) has been defined as a key criterion for determining the recovery of these species under the US Endangered Species Act (NMFS, 2015). The loss of A. palmata thicket area thus represents a trend opposing species recovery. The density of all A. palmata colonies also shows negative trends at most sites, both before and during the acute thermal bleaching event (Table 1).

Overall, population enhancement is associated with reef-scale positive trends in Acropora cervicornis in the Florida Keys, though a (order-of-magnitude) lower level of outplanting effort (predominantly during a bleaching year) did not appear to be adequate to produce a similar relationship for A. palmata. Also, positive effects of outplanting A. cervicornis appeared to be damped by a massive thermal stress event in 2014–15. If the intent is to recover these foundation species (as mandated by the Endangered Species Act) and maintain reef ecosystem function, our results point to the need for ongoing population enhancement efforts as a stop-gap strategy to prevent further population declines while the paramount need to curtail climate change is addressed (NMFS, 2015).

Supplemental Information

Supplemental Information 1 Raw data for Acropora spp. waypoints and search tracks

Click here for additional data file.

Table S1 Outplant inventory

Total numbers of outplants for Acropora cervicornis (Ac) and A. palmata (Ap) by site and year as reported by the Coral Restoration Foundation (J. Levy and K. Ripple, pers. comm).

Click here for additional data file.

Table S2 Site and date inventory

Names, abbreviations, coordinates, and census dates for each site arranged in alphabetical order.

Click here for additional data file.

Table S3 Summary for the longest interval observed at each site

Summary of congruent observed areas, colony densities, and number of outplants for each species over the longest interval of observation at each site. Change in density is represented as a proportion of the initial density. Information on numbers of outplants provided by Coral Restoration Foundation, the only organization performing large-scale population enhancement in the study area. These data were used to summarize overall trends and effect of population enhancement (see text, Table 2 and Fig. 4A).

Click here for additional data file.

Figure S1 Compiled reef census maps

Images showing the outcomes (tracks and waypoints) for each census at each site. Yellow symbols represent A. palmata colonies, purple symbols represent A. cervicornis colonies, and red symbols show the outline of A. palmata ‘thickets’ (see text for explanation) with different shaped symbols used for different years. Sites are are arranged in alphabetical order. Coordinates are given in Table S2. Base imagery of the reef from GoogleEarth.

Click here for additional data file.

Figure S2 Temporal comparison map for Grecian Rocks, 2014–2015

Observed search tracks and waypoint features mapped for Grecian Rocks reef between 2014 (A; waypoints as asterisks) and 2015 (B; waypoints as triangles). A. palmata colony waypoints are depicted in yellow, A. cervicornis colonies in purple, and A. palmata thicktet outline points in red. (C) Merged maps for temporal comparison showing the congruent observed area (determined by GIS intersect of the polygons determined by the search tracks for each year) for both years and the overlayed colony distribution observed in both years.

Click here for additional data file.

Field assistance by KL Kramer, A Valdivia, AJ Bright, RE Pausch, L Richter, and M Connelly is gratefully acknowledged. RE Pausch provided assistance with figure preparation. The Coral Restoration Foundation graciously provided information on total population enhancement effort by reef.

Additional Information and Declarations

Competing Interests

Author Contributions

Data Availability

The authors declare there are no competing interests.

Margaret W. Miller conceived and designed the experiments, performed the experiments, analyzed the data, wrote the paper, prepared figures and/or tables.

Katryna Kerr performed the experiments, analyzed the data, prepared figures and/or tables, reviewed drafts of the paper.

Dana E. Williams conceived and designed the experiments, performed the experiments, analyzed the data, prepared figures and/or tables, reviewed drafts of the paper.

The following information was supplied regarding data availability:

The raw data has been supplied as Supplemental Dataset.

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
