# Peer review of "Reef-scale trends in Florida Acropora spp. abundance and the effects of population enhancement"

_PeerJ, doi:10.7717/peerj.2523_

## Round 0.1 · original submission · Major Revisions

I have heard back from three reviewers, all of whom offered positive responses to your manuscript. All three have made some suggestions to help improve your work. While you can examine the details in their comments and attached files, two of three reviewers specifically wish to see some editing or changes in the discussion and ending of this work. As well, in order to make your work reproducible, please include more information in your M&M on colony collection etc. as recommended by reviewer 1.

While how you respond to these suggestions could result in either 'major' or 'minor' revisions, as some of the suggestions could entail larger changes, my decision is 'major revision'.

·

Basic reporting

This paper is generally well-written and based on a topic of increasing interest to the international coral community.

There are places where minor edits are required and I have highlighted those in the attached annotated pdf.

The introduction could benefit with some further referencing especially in regard to the current status of Acropora populations and the threats they face.

Some additional comments to help improve the clarity of figure and table captions is included in the annotated pdf.

It does appear that any of the raw data has been made available.

Experimental design

The methods section left me with many questions including:

1. Where were the source colonies collected from?
2. How many and how much of the wild colonies were collected?
3. How big are the fragments that were grown up in the nurseries?
4. What is the estimated cost (financial) of collecting-growing up and transplanting out a single colony?
5. Did the fragmented and transplanted colonies spawn?
6. Was the number of juveniles surveyed?

I also found it difficult to interpret if there was an overall net benefit of restoration efforts from 2005-2015. The data was presented in two batches of before and after bleaching – however this makes it tricky to interpret the net result – please add the overall decadal trend data to the relevant figures.

Validity of the findings

I am not convinced the authors have provided enough information for the reader to judge the success or otherwise of the coral restoration efforts. The discussion comes across as a little unbalanced – especially the key conclusion that continued enhancement efforts are needed.

In addition to the questions listed in the experimental design section I would also like the authors to have a go at answering the following questions - if they can answer these it may help to balance out the discussion.:

7. How does this result in the upper Florida Keys relate to results in other parts of the Florida tract?
8. If climate change is not curtailed do the authors really recommend this as a feasible recovery action?
9. Are the observed benefits reasonable in face of the costs?
10. Do the authors have any insight into why palmata has not responded to the out-planting?
10. Is the legal mandate to 'recover' imperiled species reasonable?

I would also like the authors to reconsider their final sentence which refers to identifying thermally resistant traits – this is a controversial and untested manipulative approach to coral conservation and quite separate to the topic of this manuscript. I would urge the authors to consider ending their manuscript with a targeted statement reinforcing the need to curtail climate change as their manuscript shows this is the key factor that threatens to derail the success of coral restoration efforts.

Additional comments

This is an interesting and topical paper - however further details are needed to help convince the readers that the costs of this approach outweigh the benefits. I have highlighted key areas that need attention in the annotated pdf that is attached.
Regards Zoe

·

Basic reporting

The article meets all of the required standards

Experimental design

No comments besides those that appear in the General Comments

Validity of the findings

No comments besides those that appear in the detailed General Comments

Additional comments

Please see my detailed comments on the attached file

Reviewer 3 ·

Basic reporting

This article evaluated changes to Acropora spp populations across broad reef areas in the Florida Keys and measured the effects of restoration efforts on colony density and thicket area through time. This is an intriguing idea and is very pertinent given the dynamic nature of the Caribbean Acropora spp. It is very difficult to get good population estimates over small spatial scale, especially now that the populations have dwindled. Larger scale efforts are needed to improve population estimates and study the corals’ dynamic nature. The paper started off strong with a good intro but discussion had some issues that should be addressed before publication.

The discussion requires more effort. It didn’t offer much and didn’t relate the findings to any other work on Acerv and/or Apalm in the FL keys or the broader Caribbean. Much more work went into the intro in this regard than the discussion. It appears to be written in haste to push out a publication without taking the time to really contribute to the overall state of restoration research. For example, the paper speculates about the causes of Acerv decline in recent years, but doesn’t point to any other research to help substantiate the idea. The authors should use the same approach as their intro and relate their findings to the cited works of previous and ongoing studies.

Experimental design

The authors need to define a thicket better. Line 121 only states “where it was not feasible to demarcate individual colonies”. This is too ambiguous, especially when it is used to show a decline in the population via thicket area. What were the criteria used to decide to map individual corals vs a group? How were the thickets included in the density analyses? It is important to understand how the thicket areas were drawn and the density was calculated. It is especially important in this case given there were only two study periods and multiple divers. There is no indication of variance in the measurements between divers or with the same diver so any information that supports this variance did not contribute significantly to the differences found, the better. This directly affects Fig 2 and the main outcomes of this study. Luckily the differences are stark and there are probably other studies that support their outcomes, which they should incorporate in the discussion.

Validity of the findings

One point the discussion really missed out on was the broader scale aspect of Acropora population estimates. These are very important and relevant and should be given a proper discussion to address how the estimates were better at the broader scale than at a smaller level. There is another Peer J manuscript in late review stages by D’Antonio et al. that could be of use for this paper’s discussion. Perhaps it can be shared with the authors.

Additional comments

Specific edits
Line 49: Correct sentence
Line 214: change “overcome even the enhancement effort and yielded” to “adversely affected colony density because”
Line 215 change “in spite of” to “despite”.
Line 215: It seems there would be some other reference info on the fate of these colonies instead of speculation. Have any other studies in the region showed similar data for the timeframe? Does CRF monitor their outplants? Perhaps they have more info to substantiate this.
Line 221: It’s difficult to associate a resolution to the survey design in this sense. Change “The resolution of our survey technique was low” to “The precision of our survey estimates varied”
Line 75: Change “low-resolution” to “low precision”
Lines 240-243: Surely there are other studies that can be cited to support these findings.
Fig 1. The thin white line tracks are too hard to see against the light image. The orange polygons are also hard to see against the brown reef. Use colors that make the mapped coral more obvious. Change the color so they are more visible. The figure is not designed well with different sized boxes and images at different scales with no scale bars. It would be best if the image matched the timeframe instead of using the same image for each scene. D and E could have the 2014 and 2015 image respectively.
I think including figures for all the site maps as supplementary material would be valuable information.
Figure 4 A and B have inconsistent labeling, fonts and capitals.
Table 1. explain early and late labeling

---

## Round 0.2 · Minor Revisions

I have heard back from one reviewer, who has one remaining point to consider. Thus, this is a quite 'minor revision'.

Reviewer 3 ·

Basic reporting

The authors have done a nice job revising the manuscript.

Experimental design

The authors have done a nice job revising the manuscript.

Validity of the findings

The authors have done a nice job revising the manuscript.

Additional comments

I have no issue with this paper's main message that outplanting increases number of colonies at a large scale and the authors did a nice job on the revisions. One last thing that concerns me is that the site locations of the study directly relate to their conclusions. The paper plainly states the study is limited to the Florida Keys and gives a good indication of the state of Acropora in the Florida Keys. However, a large portion of the reef tract that continues north along SE FL is ignored. This area has had substantial A. cervicornis populations documented (Vargas-Angel et al. 2003, Walker et al. 2012, and D'Antonio et al. 2016). For this reason, I think it's important that the authors better-define the geographic boundary of their study area (at least the northern boundary) and acknowledge their study does not include other known areas northward along the same reef tract that have substantially larger natural populations of A cervicornis "thickets".

---

## Round 0.3 · accepted · Accept

Thank you for attending to these small details. I also could not find the D'Antonio et al. 2016 paper, but as you already have some references, this should be fine.

I look forward to seeing your paper in published form.